# Grazing Intensities Regulated the Effects of Seasonal Dietary Pattern on Gut Bacterial Community Composition of Sheep

**DOI:** 10.3390/microorganisms13061392

**Published:** 2025-06-14

**Authors:** Pengzhen Li, Zhenhao Zhang, Thomas A. Monaco, Yao Dong, Yuping Rong

**Affiliations:** 1Department of Grassland Resource and Ecology, College of Grassland Science and Technology, China Agricultural University, Beijing 100193, China; b20213241028@cau.edu.cn (P.L.); b20243241309@cau.edu.cn (Z.Z.); s20223243474@cau.edu.cn (Y.D.); 2U.S. Department of Agriculture, Agricultural Research Service, Forage and Range Research Laboratory, Utah State University, Logan, UT 84322-6300, USA; tom.monaco@usda.gov

**Keywords:** enterotype, seasonal diet, 16S rRNA sequencing, lamb

## Abstract

Gut microbiota “enterotypes” are strongly associated with diet and host health. For grazing animals, plant species richness and nutrient content of vegetation may alter the food supply and diet composition of animals. Understanding this relationship is critical to clarify the adaption of gut microbiota to changes in vegetation quantity and quality in grassland ecosystems. Here, we studied the relationship between dietary and gut microbiota composition of sheep (lambs) over a growing season in a grassland ecosystem in northern China. Variation in vegetation composition among grazing intensities was greatest in September: and sheep preferred forbs and Rosaceae throughout the grazing period in all grazing treatments, yet their preference for Fabaceae was reduced in HG treatments in September. Grazing intensity and seasonal variations in food resource availability influenced dietary patterns, which in turn affected gut bacterial community composition. Enterotype 1, dominated by *Christensenellaceae_R_7_group* and *Clostridia_UCG_014_unclassified*, predominated during the warm season (July) for both LG and HG treatments. In contrast, Enterotype 2, dominated by *Escherichia_Shigella*, prevailed during the cool season (September) in HG. Diversity of Enterotype 1 exceeded (*p* < 0.001) that of Enterotype 2. For MG, Enterotype 1 and Enterotype 2 were evenly distributed over the grazing period. Our results highlight the importance of regulating grazing intensity to maintain the balance and health of gut microbiota according to temporal changes in plant nutrients and aboveground biomass of grassland ecosystems.

## 1. Introduction

Mammalian guts harbor trillions of microbes, which play important roles in host biology, including nutrition, immune system development, and behavior [1]. Changes in gut microbial composition have been linked to host health and disease [2,3]. Gut microbial communities can be classified into several stable “enterotypes,” which are defined by the relative abundance of key bacterial taxa and shaped by diet and their resilience to short-term perturbations caused by environmental changes [4]. Recent work suggests that seasonal variations in animal gut bacterial community are driven by dietary shifts, yet the influence of fluctuating vegetation on this relationship is less understood [5]. Because grazing animals in managed pastures and grassland ecosystems are challenged with obtaining sufficient nutrition in the face of uneven distribution of plant species, fluctuating vegetation productivity, and forage nutrient levels [6], understanding the factors controlling enterotype dynamics could potentially contribute to improved animal health management.

Seasonal dietary composition is considered a key factor shaping gut microbiota in animals [7], and seasonal changes in gut microbes are closely related to nutrient composition of vegetation [8]. Grazing can also strongly influence above- and belowground biomass production, root/shoot ratios, vegetation composition, and species richness and evenness [9,10]. Consequently, animal health and forage supply depend on grazing intensity [11]. For example, heavy grazing can constrain forage resource availability and alter forage quality, which can impose foraging stress on animals and force them to alter their dietary consumption [12]. When the supply of high-quality forage becomes limiting, or if plant species diversity declines during seasonal changes in vegetation, animals are forced to choose lower-quality forage [13]. Accordingly, the reduction in species diversity under heavy grazing pressure can reduce animal foraging selectivity and modify their foraging behavior [14].

Because animals rely heavily on a diverse array of microbes to digest plants, gut microbes are closely related to foraging behavior [15]. The response of gut microbiota to seasonal forage supply has been studied for yak (*Bos grunniens*) [16], African buffalo (*Syncerus caffer*) [4], North American bison (*Bison bison*) [17], sheep (*Ovis aries*) [18] and goat (*Capra hircus*) [19]. In young goats, two enterotypes exhibited differences in microbial functions related to growth rate, amino acid synthesis, and starch metabolism [19]. In African buffalos, gut enterotypes adapted to temporal variation in food availability [4]. However, most studies on gut microbiota dynamics have not fully considered how shifting forage availability impacts gut microbiota. For example, it is uncertain how grazing intensity, which is known to modify forage composition and abundance [10], can influence diet preference in sheep. Thus, a clearer understanding of how grazing intensity affects forage availability and the gut microbiota in sheep can lead to more effective management strategies for livestock health and productivity.

Temperate grasslands of northern China are important ecosystems where sheep production and ecological integrity must be balanced through effective grazing management. This study aimed to (1) determine the seasonal dietary patterns and their impact on gut bacterial community composition; (2) explore how grazing intensity moderates the relationship between diet and gut microbiota; and (3) examine the effects of grazing intensity and month of season on gut bacterial community function. Addressing these objectives will help clarify the temporal dynamics of gut microbiota and dietary changes over the growing season and offer new insights into how grazing intensity interacts with seasonal variations to shape gut microbiota composition in sheep.

## 2. Materials and Methods

### 2.1. Site Description

Research was conducted at the Grassland Ecosystem Research Station (44°00′ N, 116°26′ E, elevation 1150 m), Xilinhot, Inner Mongolia, China (Figure 1). The site is a temperate grassland with annual mean temperature ranging from −1.8 to 1.6 °C, mean minimum temperature in January (−22.0 °C), and mean maximum temperature in July (20.1 °C). Annual mean precipitation ranges from 170 to 370 mm, with about 80% occurring during the growing season between June and August. Vegetation is dominated by two perennial grasses (*Leymus chinensis* [Trin.]) Tzvelev. and *Stipa grandis* P. Smirn.) and to a lesser extent the perennial grasses (*Cleistogenes squarrosa* (L.) Schult., *Agropyron cristatum* (L.) Gaertn.), the sedge (*Carex korshinskyi* Kom.), and several perennial herbs (*Sibbaldianthe bifurca* (L.) D.F. Austin., *Artemisia frigida* Willd., *Gagea lutea* (L.) Ker Gawl., *Astragalus laxmannii* Fisch. ex DC., *Iris tenuifolia* Pall.). Total vegetation cover is typically 40–50% with a mean height of 10–15 cm. Soils are classified as dark chestnut soil (Calcic Chernozem according to ISSS Working Group RB, 1998), with a loamy sand texture [12].

### 2.2. Grazing Experiment

From a broader set of treatments associated with a prior study [20], our current experiment focused on three treatments that varied in grazing intensity. Light, moderate, and heavy grazing treatments were applied to 2 ha plots, each replicated three times, and contained either 6 (0.50 sheep units·ha^−1^·yr^−1^), 12 (0.96 sheep units·ha^−1^·yr^−1^), or 17 (1.38 sheep units·ha^−1^·yr^−1^), respectively. Lambs continuously grazed during the growing season (15 June–15 September) in 2022 and 2023, and all measurements were made in 2023. Stocking rates were calculated based on the total liveweight of lambs in each plot (measured at the beginning of grazing and monthly thereafter) and converted into sheep units using a 50 kg adult sheep equivalent. Animals were local-breed Ujimqin male lambs, with an average liveweight of 25 ± 2.5 kg. Lambs were orally administered an anthelmintic to treat internal and external worm infections, ear-tagged for identification, and allocated randomly to the plots. Lambs grazed throughout the day and had access to a shelter in each plot for rest. Animals had free access to water and a mineral block, but no additional supplements were given during the grazing periods.

### 2.3. Vegetation Sampling and Nutritional Analyses

Vegetation samples were collected on 14 July, 15 August, and 14 September in 2023. Ten quadrats (50 cm × 50 cm) were randomly placed in each plot and the current year’s aboveground biomass was cut to ground level. If the collected biomass of certain plant groups was insufficient for forage quality analysis, 5–10 additional quadrats were established within each plot to obtain adequate plant material. All plant species were identified to analyze differences in vegetation across the growing season within each grazing treatment. In addition, species were grouped into one of six families (Poaceae, Asteraceae, Cyperaceae, Liliaceae, Rosaceae, and Fabaceae) or into a general category of forbs. Vegetation samples were dried in an oven at 60 °C for 48 h to determine vegetation dry matter and ground using a KF-25 grinder (Kangyuanxin, Wenzhou, China) to pass a 1 mm screen for subsequent forage quality determination [21], including crude protein (CP), neutral detergent fiber (NDF), and acid detergent fiber (ADF). Crude protein was determined using the Kjeldahl digestion method (Kjeltec 2300, FOSS Analytical AB, Hoganas, Sweden). The NDF and ADF were measured following Van Soest’s method [22], incorporating heat-stable alpha-amylase and sodium sulfite in the NDF procedure.

### 2.4. Diet Composition–DNA Metabarcoding Sequence Analysis

Feces were sampled via rectal grab sampling from three lambs from each plot of LG, MG and HG on five consecutive days in mid-July, mid-August, and mid-September. Samples over the five days for each lamb were thoroughly mixed, placed in unused freezer tubes, and immediately stored in liquid nitrogen containers, yielding a total of 81 samples (3 treatments × 3 plots × 3 lambs × 3 sampling periods) for subsequent diet and microbiota DNA analyses.

Diet composition was quantified using fecal DNA metabarcoding. The DNA was extracted from a 0.2 g feces sample with QIAamp^®^ Fast DNA Stool Mini Kit (QIAGEN, Hilden, Germany) and an extraction blank was processed to monitor cross-contamination. DNA was quantified using NanoDrop-2000 UV-Vis Spectro-photometer (Thermo Fisher Scientific, Wilmington, DE, USA).

The chloroplast *rbcL* gene was used for DNA metabarcoding sequencing, with primers *rbcL* P1-F (CTTACCAGYCTTGATCGTTACAAAGG) and *rbcL* P1-R (GTAAAATCAAGTCCACCRCG) [23,24]. For PCR assays, 10 μL reaction mixtures were prepared for each sample, including 0.3 μL of each primer, 0.2 μL KOD FX Neo polymerase, 2 μL dNTPs, 5 μL KOD FX Neo buffer, and 50 ng of DNA template. The thermal cycling program consisted of initial denaturation at 95 °C for 5 min, followed by 40 cycles of denaturation at 94 °C for 20 s, annealing at 55 °C for 30 s, extension at 72 °C for 1 min, and a final extension at 72 °C for 5 min. All PCRs included a no-template negative control. Each primer was labeled with a 16–nucleotide multiplex identifier tag at the 5′ end, differing from the other tag by 8 nucleotides, allowing for unique labeling of PCR products. Sequencing was performed using a HiSeq 2500 platform (Illumina Inc., San Diego, CA, USA).

Sequence quality control and preliminary identification were conducted by Trimmomatic v0.33 (Usadel Lab, Aachen, Germany), where sequences with an average Illumina fastq quality score <20 were not considered. Cutadapt software (v1.8.3, TU Dortmund University, DE, Germany, and National Bioinformatics Infrastructure Sweden, SE, Sweden) was used to identify and remove primer sequences, with parameters allowing a maximum mismatch rate of 20% and a minimum coverage of 80% for primer sequence identification. The processed data were denoised using DADA2 method in QIIME2 [25] (version 2020.6). Quality control of data was further assessed using the FilterAndTrim function with maxEE to 2 (where EE = sum(10^(−Q/10))). We also applied the learnErrors function to build model, dada function for denoising, and the mergePairs function for merging paired end reads. Identical sequences were collapsed, and plant species were assigned to reference sequences based on their unique matches (100% identity) with DNA metabarcoding sequences. Only unique sequences with 100% identity to reference sequences were retained for further analysis. The most refined taxonomic assignments were made using the Naive Bayes classifier with the Fungene and NCBI databases. The summarize_taxa command was used to group identical sequences, perform within-sample statistics, and quantify the relative read abundance of each sequence. The proportion of each Amplicon Sequence Variant (ASV) in each sample represents the relative abundance of the components of animal’s diet. This method has been widely applied to quantify the proportions of consumed plant species in animal dietary studies and has been validated across multiple contexts [6,16,26].

### 2.5. Gut Microbiota Composition–16S rRNA Gene Illumina Sequencing and Enterotype Clustering

The V3-V4 region of the 16S rRNA gene was sequenced on NovaSeq 6000 platform (Illumina Inc., San Diego, CA, USA) with primers (341F/806R). For PCR assays, 50 μL of each of the 30 ng DNA template, fusion primer, and PCR master mix were mixed. The PCR cycles started with a 5 min denaturation at 95 °C, followed by 30 cycles each consisting of 95 °C for 30 s, 50 °C for 30 s, 72 °C for 40 s, and followed by a final step of 72 °C for 7 min. Polymerase Chain Reaction (PCR) products were purified with VAHTS DNA Clean Beads (Vazyme Biotech, Nanjing, China) and eluted in elution buffer. Libraries were qualified by the Agilent 2100 bioanalyzer (Agilent, Santa Clara, CA, USA). The amplicons were sequenced on NovaSeq 6000 (Illumina, San Diego, CA, USA) and generated 2 × 250 bp paired-end reads. Trimmomatic v0.33 was used to filter the raw reads obtained from sequencing, followed using Cutadapt 1.9.1 to identify and remove primer sequences. Denoising was performed using the DADA2 method in QIIME 2 (2020.6) [25], during which paired-end sequences were merged, and chimeric sequences were removed, resulting in non-chimeric reads. After denoising with DADA2, singleton ASVs were removed during the subsequent quality filtering step. Subsequently, ASVs were aligned to the reference database using blastn(v2.9.0, NCBI, Bethesda, MD, USA), and any ASVs that could not be accurately matched to the reference database were classified using the classify-sklearn classifier as a supplement.

The robustness of clusters was assessed by the Calinski–Harabasz (CH) index and silhouette score. To identify genus taxa contributing to enterotype groups based on Bray–Curtis dissimilarity and partitioning around medoid (PAM) in genus-level abundance, we applied the SIMPER method, which identifies genus taxa contributing to similarity within and dissimilarity between enterotypes and ranks their contribution.

### 2.6. Vegetation Composition and Diet Selectivity

The proportion of each plant family to total above-ground biomass was quantified, representing the relative abundance of vegetation components and a quantitative measure of food availability. These data were combined with the family diet proportions determined from fecal analysis to assess diet selectivity by applying Jacobs’ index (*D*) [27], in which the utilization of a food item (its proportion in diet, *r*) is related to its relative availability in vegetation (*p*) according to following equation:D (r−p)∕(r+p−2rp )

The selectivity index varied from −1 to 0 for negative selection (“avoidance”) and from 0 to 1 for positive selection (“preference”): a value of 0 corresponds to utilization of a food item in proportion to its relative availability.

### 2.7. Statistical Analysis

We used standard R commands to generate relative abundance values for composition of vegetation, sheep diets, and microbiota of sheep guts. Analysis of variance (ANOVA) was used to assess the effects of grazing intensity, month of season, and their interaction on these responses as forage quality. We used nonmetric multidimensional scaling (NMDS) and PERMANOVA within the *vegan* package to compare differences in diet and microbiota composition across grazing intensity and season and analysis a season of month x grazing level interaction. To account for the non-normal distribution of selectivity data, we conducted Wilcoxon tests to evaluate whether Jacobs’ *D* for forages differed significantly from zero. These statistical analyses were conducted using SPSS 22.0 (IBM Corp., Armonk, NY, USA).

Based on reports of functional differences in enterotypes, we explored whether gut microbiota of sheep shaped clusters with distinct functional characteristics. For enterotype comparisons, samples were pooled into bins (July, August, and September), and significance among months was identified using Fisher’s exact test with false discovery rate (FDR) correction of *p*-values. False discovery rate was applied at a level of α = 0.05 per tested correlation and significance for multiple comparisons using the stats package (v4.3.0). We calculated differences in Shannon–Weiner index, relative abundance of major bacterial taxa, and functional pathways between two enterotypes using the vegan (v2.6.4) package. To identify genus taxa contributing to differences between enterotype groups based on Bray–Curtis dissimilarity, we applied the SIMPER method, which identifies and ranks the genus taxa contributing most to the dissimilarity between enterotypes.

Linear discriminant analysis (LDA) effect sizes (LEfSe) were calculated to analyze significant differences in the abundance of specific groups between enterotypes (defined as Enterotype 1 or Enterotype 2). LEfSe combines the Kruskal–Wallis test and Wilcoxon test to identify features with significant differences within defined groups, and LDA was used to evaluate the effect size of each feature [28]. Microbial function analysis was performed using PICRUSt3 based on ASVs clustered from 16S rRNA sequencing data, then metabolic predictions were identified from the Kyoto Encyclopedia of Genes and Genomes (KEGG) database. Differences in predicted results were processed using *t*-test analyses with the FDR correction [8]. *p*-values for multiple comparisons were adjusted with the Bonferroni method. All analyses were conducted in R version 3.6.1.

## 3. Results

### 3.1. Gut Microbiota and Diet Composition

Based on PERMANOVA, compositions of both gut microbiota and sheep diets were significantly influenced by grazing intensity, season month, and their interaction (*p* < 0.001; Figure 2a,b). Regarding the gut microbiota, Prevotellaceae, Ruminococcaceae, Bacteroidaceae, Lachnospiraceae, Enterobacteriaceae, and Christensenellaceae showed the strongest associations with the ordination. Among the vegetation variables related to dietary composition, species belonging to Poaceae, Cyperaceae, Rosaceae, and various forbs exhibited the most significant contributions to the ordination.

### 3.2. Aboveground Biomass and Nutrient Composition

No differences were found in aboveground biomass among grazing treatments in July, yet Poaceae values were lower for HG than LG in August and lower than both LG and MG in September (Appendix A; *p* < 0.05). In September, aboveground biomass for Liliaceae and Fabaceae were also lower in HG and MG than LG, respectively (*p* < 0.05). Few differences were found among grazing levels and months of the season for forage quality variables, and when differences occurred, they were not consistent within plant family groups (Appendix A).

### 3.3. Relative Abundance of Plant Families in Vegetation and Diets

Vegetation over the growing season was clearly dominated by Poaceae for all three grazing levels (Figure 3a). Averaged over the season, Poaceae values for HG were higher than in LG (Figure 3a; *p* < 0.05) and the proportion of Fabaceae and forbs for LG were higher than in MG (*p* < 0.05).

Averaging over the season, grazing treatments altered the composition of sheep diets. For example, the proportions of Poaceae in diets were higher in HG (55%) compared to LG (31%) (Figure 3b; *p* < 0.05). The proportion of Rosaceae in LG exceeded values in MG and HG (*p* < 0.05). Relative abundances of forbs in LG and MG also exceeded HG (*p* < 0.05). Months of season also influenced sheep diets. Asteraceae values for August were higher than July and September (*p* < 0.05). The proportion of Rosaceae was greater in August than July and September (*p* < 0.05). In September, Poaceae values were higher than in July and August (*p* < 0.05).

### 3.4. Selectivity for Plant Families

For all months of the season and grazing levels, sheep consistently avoided Poaceae and preferred Rosaceae (Figure 3c–e; *p* < 0.01). Forbs were also preferred across the season, yet more so for LG and MG (*p* < 0.05). Selectivity for Fabaceae shifted from preference under MG in July (*p* < 0.001) to avoidance under HG in September (*p* < 0.001). Lastly, sheep avoided Liliaceae under LG in August and Asteraceae under HG in September (*p* < 0.01).

### 3.5. Relative Abundance of Gut Microbiota

Grazing intensity altered gut microbiota composition of sheep. For example, the proportion of *Clostridia_UCG_014_unclassified* in LG was higher than MG and HG (Figure 4; *p* < 0.01). In addition, *UCG_010_unclassified* values in MG were higher than LG and HG (*p* < 0.05). The proportion of *[Eubacterium]_coprostanoligenes_group* was higher in LG than HG (*p* < 0.05). Months of season also influenced gut microbiota composition. The proportion of *Escherichia_Shigella* in September exceeded values in July and August (*p* < 0.01). In July, *Clostridia_UCG_014_unclassified* values were higher than in August and September (*p* < 0.01). The proportion of *Lachnospiraceae_unclassified* in July was higher than in September (*p* < 0.05).

### 3.6. Relationship Between Dietary and Gut Microbiota Composition

Gut samples formed two distinct enterotype clusters based on Bray–Curtis dissimilarity. Each cluster was driven by variation of its representative genera level: *Clostridia_UCG_014_unclassified* and *Christensenellaceae_R_7_group* in Enterotype 1 and *Escherichia_Shigella* in Enterotype 2 (Figure 5a). Shannon–Weiner diversity index of Enterotype 1 was significantly higher than that of Enterotype 2 (Figure 5b; *p* < 0.001). The distribution of enterotypes showed temporal variation across months under LG and HG, but not MG (Figure 5c–e). In LG and HG sites, Enterotype 1 predominated in July (*p* < 0.001), while Enterotype 2 dominated in September for HG (*p* < 0.05).

Linear discriminant Effect Size (LEfSe) analysis revealed significant differences in gut microbial composition between enterotypes (Figure 6). At the phylum level, Enterotype 1 was enriched with members of Firmicutes, while Enterotype 2 was dominated by Proteobacteria. At a lower phylogenetic level, Enterotype 1 was enriched with 19 discriminative lineages belonging to Firmicutes, 1 lineage belonging to Actinobacteria, and 2 lineages belonging to Bacteroidetes. Enterotype 2 was primarily enriched with lineages belonging to Proteobacteria at a lower phylogenetic level (Figure 6a). The genus *Bacteroides*, family *Bacteroidaceae* (all within Bacteroidetes), was found to be more abundant in Enterotype 1. Similarly, *Christensenellaceae_R_7_group*, *Clostridia_UCG_014_unclassified*, and *UCG_005* (all within Firmicutes) were also identified as more abundant in Enterotype 1. In contrast, members of the genus *Escherichia_Shigella*, family *Enterobacteriaceae* (all within Proteobacteria), were found to be more abundant in that of Enterotype 2 (Figure 6b).

### 3.7. Functional Characterization of Gut Microbiota

Relative abundances of KEGG orthologs revealed distinct functional differences between enterotypes and among grazing intensities (Figure 7a,d). Metabolic pathways and amino acid biosynthesis were significantly higher in Enterotype 1 than 2 (Figure 7b,c; *p* < 0.05). Metabolic pathways in MG exceeded values in HG (Figure 7e; *p* < 0.05), while biosynthesis of amino acids in LG exceeded HG (Figure 7f; *p* < 0.05).

## 4. Discussion

Our quantitative analysis found strong influences of grazing intensity and month of season on dietary composition of sheep. Results supported the hypothesis that grazing intensity interacts with seasonal dietary variations to influence gut microbiota. Heavy grazing reduced diet quality and influenced the composition and function of gut microbes. In contrast, under light and moderate grazing, a broader range of vegetation resources provided sheep with more opportunities to consume nutrient-rich and diverse plant species, thereby enhancing gut microbial diversity and optimizing metabolic function.

Grazing intensity is often elevated to accommodate livestock growth demands in grassland ecosystems, but its interaction with seasonal dietary variation and its effects on gut microbiota are frequently overlooked [29]. While grazing intensity is known to influence plant composition and forage availability [30], our results highlight that it also regulates how seasonal dietary shifts shape gut microbiota composition in sheep. Notable shifts in dietary preferences over the season included greater consumption of forbs under light and moderate grazing intensities, while sheep relied more on Poaceae under heavy grazing, particularly in September (Figure 3). Although Poaceae represented 84–92% of the available biomass, sheep exhibited a clear preference for forbs, suggesting a selective foraging strategy. Spitzer et al. [26] reported that animals forage selectively to meet their nutritional needs. Similarly, our findings suggest that sheep preferred nutrient-rich forbs over the dominant Poaceae, likely to optimize nutrient intake despite the lower abundance of forbs. Importantly, seasonal dietary shifts varied across grazing intensities—sheep under light and moderate grazing had more consistent access to forbs, while those under heavy grazing increasingly relied on Poaceae later in the season. In addition, seasonal changes in plant secondary metabolites or physical traits such as increased lignification may also have contributed to the observed dietary shifts [31]. This is consistent with our observation of a significant decline in the selection of Fabaceae by sheep in September (Figure 3), suggesting possible avoidance due to reduced palatability or increased antinutritive compounds. These dietary changes, driven by seasonal variation in plant nutrient content and availability, subsequently influenced gut microbiota composition. These results further supported our hypothesis that grazing intensity alters the effect of seasonal dietary variation on gut microbiota. The microbial shifts observed in sheep guts align with previous studies [11,32], suggesting that diet is a primary driver of microbial gut composition, but our findings further emphasize that the extent of these microbial changes depend on grazing intensity. Such effects may be attributed to differences in forage availability, leading to gut microbiota reassembly in response to seasonal dietary shifts under different grazing intensities [33]. Our results underscore the importance of grazing management in regulating the seasonal dynamics of diet and gut microbiota, which are critical for maintaining animal health and ecosystem sustainability.

This study provides biological insights into how grazing intensity regulates the seasonal dynamics of gut enterotypes and their functional implications. We found that enterotype transitions in sheep were strongly influenced by seasonal dietary variations, with grazing intensity further shaping these effects. Changes in plant nutritional composition, driven by both grazing intensity and month of season, were closely associated with shifts in enterotype distribution (Figure 5). Wu et al. [34] reported that plant protein and fiber content influenced enterotype transitions in young goats, which parallels our findings. However, unlike the stable enterotype distribution observed in African buffalo under resource-rich diets [4], our study revealed pronounced seasonal shifts in enterotypes, particularly under both light and heavy grazing. Specifically, Enterotype 1 was dominant in July under both light and heavy grazing, while that of Enterotype 2 became predominant in September under heavy grazing (Figure 5), suggesting that seasonal dietary shifts interact with grazing intensity to influence gut microbiota composition.

Our results indicate that grazing intensity modulates the seasonal effects on enterotype distribution, leading to distinct microbial composition and functional differences (Figure 6 and Figure 7). Enterotype 1 exhibited higher microbial diversity, along with enriched metabolic pathways and amino acid biosynthesis (Figure 5 and Figure 7). Previous studies have shown that animals rely on diverse gut microbial communities to optimize energy extraction [15]. Consistent with this, our study suggests that under conditions of abundant vegetation, higher microbial diversity enhances the metabolic potential of sheep guts. In contrast, a reduction in gut microbial diversity, particularly under heavy grazing, disrupted bacterial community stability and potentially compromised animal health as seen in a previous study [4]. However, Wang et al. [19] found that Enterotype 1 in young goats had lower microbial diversity but higher energy conversion, which may be in the face of differences in animal breeds and vegetation composition.

In July, Enterotype 1 dominated in light and heavy grazing treatments, and were characterized by a high abundance of Firmicutes, particularly *Christensenellaceae_R_7_group*, which is associated with host nutrient utilization and lipid metabolism [35]. By September, Enterotype 2 became dominant under heavy grazing, featuring a high abundance of Proteobacteria, particularly *Escherichia_Shigella* (Figure 6). The proliferation of *Escherichia_Shigella* in September suggests a compromised bacterial community stability, as this genus has been linked to intestinal dysbiosis [36]. The functional consequences of these microbial shifts were evident in the reduced microbial metabolic potential of Enterotype 2 compared to Enterotype 1. Additionally, amino acid biosynthesis pathways in gut microbes were significantly lower in HG than in LG, and the metabolic pathways were also lower in HG compared to MG (Figure 7), indicating potential bacterial community dysregulation and increased disease risk [37].

These findings suggest that grazing intensity regulates the effects of seasonal dietary changes on gut microbiota composition and function. Enhancing grassland productivity and maintaining the diversity of natural pastures is crucial to improving the efficiency of livestock production and increasing the income of herders. However, our study has certain limitations. It was conducted in a temperate grassland region, focused solely on lambs, and spanned one year. Thus, further exploration would be valuable to assess whether these findings are applicable to studies involving adult animals, as well as to long-term, multi-species research, and whether they can be generalized to other ecosystems. Additionally, given the critical role of rumen microbiota in forage digestion and animal nutrition, future research should incorporate rumen microbial analysis to provide a more comprehensive understanding of diet–microbiome–host interactions in grazing systems.

## 5. Conclusions

Our study demonstrates that grazing intensity plays a crucial role in regulating the effects of seasonal dietary changes on the gut microbiota composition of sheep. We found that seasonal shifts in plant nutritional content, combined with different grazing intensities, significantly influenced enterotype transitions and microbial diversity. These findings underscore the importance of adaptive grazing management in maintaining a balanced gut bacterial community, which is vital for livestock health and ecosystem sustainability. Reducing grazing intensity could improve forage quality, stabilize microbial diversity, and enhance metabolic function, offering valuable insights for sustainable livestock management in grassland ecosystems.

## Figures and Tables

**Figure 1 microorganisms-13-01392-f001:**
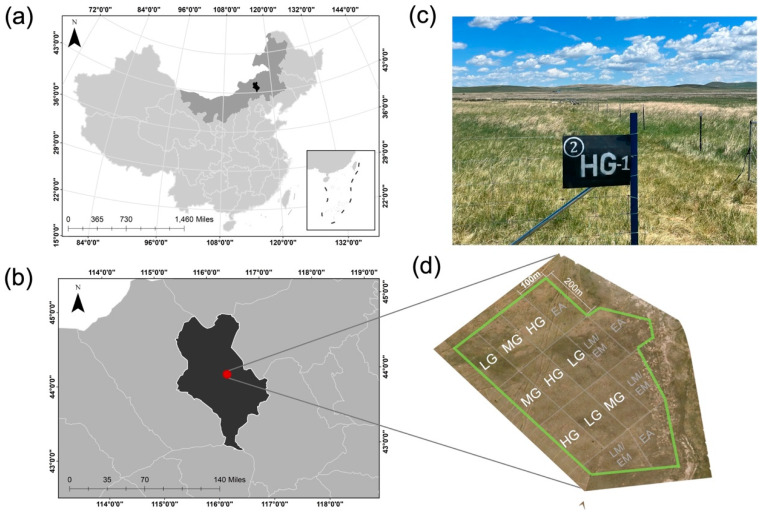
Map of study site in Xilinhot, Inner Mongolia Autonomous Region (**a**,**b**). Photographs of experiment plot and layout (**c**,**d**), where the nine plots included in this study are indicated with white letters and unused plots are shown with gray letters. Sheep grazing intensities included light (LG), moderate (MG), and heavy (HG).

**Figure 2 microorganisms-13-01392-f002:**
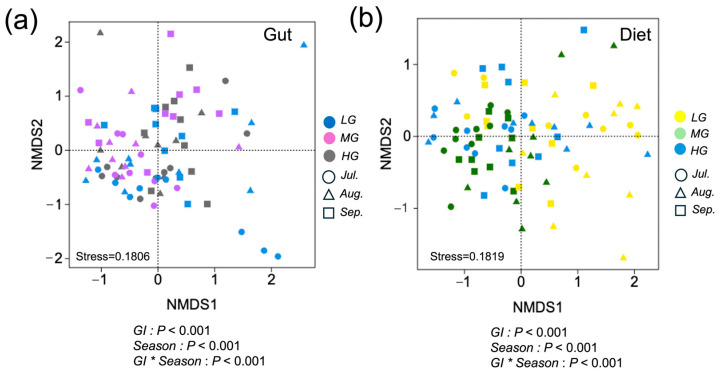
Sheep diet and gut microbiota composition differentiation among grazing intensity and seasonal month, and the interaction of grazing intensity and season (designations follow Figure 1) illustrated by nonmetric multidimensional scaling (NMDS) ordination. Solid circles represent individual gut microbiota distinguished with different grazing intensity for LG (blue), MG (purple), and HG (gray). Different shapes represent individual gut microbiota samples for July (circle), August (triangle), and HG (square) (**a**). Solid circles represent individual dietary samples for LG (yellow), MG (green), and HG (blue). Different shapes represent individual dietary samples for July (circle), August (triangle), and September (square) (**b**). Closer sample positions within the ordination indicate more similar diets and gut microbiota. *p*-values indicate results of PERMANOVA analyses.

**Figure 3 microorganisms-13-01392-f003:**
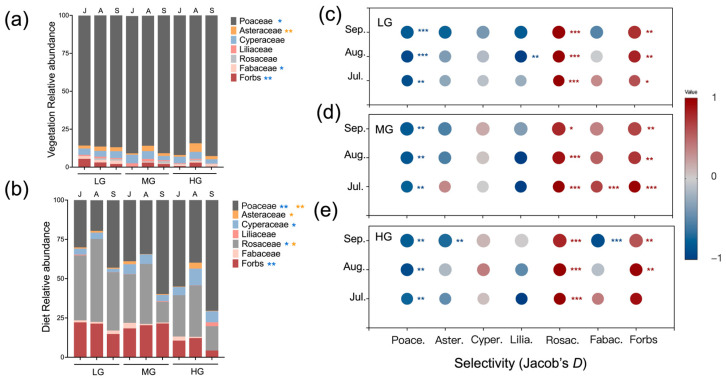
Mean relative abundance of plant families and select forbs within vegetation (**a**) and sheep diets (**b**) for seasonal months and grazing intensities (designations follow Figure 1). Mean diet selectivity was based on Jacob’s D index for seasonal months in LG (**c**), MG (**d**) and HG (**e**). Relative abundance ranged from 1 to −1. Diet proportions lower than vegetation proportions indicate avoidance (shown in blue), yet the opposite indicates preference (shown in red). Color shading indicates degree of avoidance/preference and gray symbols indicate when the proportions of diet and vegetation are the same. Seasonal months included July (J), August (A), and September (S). Asterisks indicate significant differences in vegetation and diet composition among seasonal months (yellow) or grazing levels (blue) (* *p* < 0.05, ** *p* < 0.01, *** *p* < 0.001).

**Figure 4 microorganisms-13-01392-f004:**
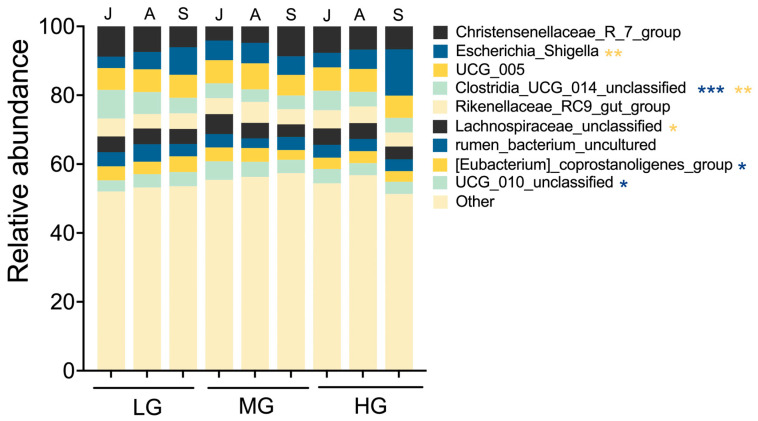
Microbiota composition of sheep guts at the genus level for seasonal month under different grazing intensities (designations follow Figure 1 and Figure 3). Values are relative abundances of the 10 most abundant genera aggregated and colored for each seasonal month. Low abundance taxa are grouped together as “others.” This low-abundance group accounted for 53.9% of total reads, with 29.1% classified to genus level and the remainder assigned to family or higher taxonomic ranks. Asterisks indicate significant differences in microbiota composition among season months (yellow) or grazing levels (blue) (* *p* < 0.05, ** *p* < 0.01, *** *p* < 0.001).

**Figure 5 microorganisms-13-01392-f005:**
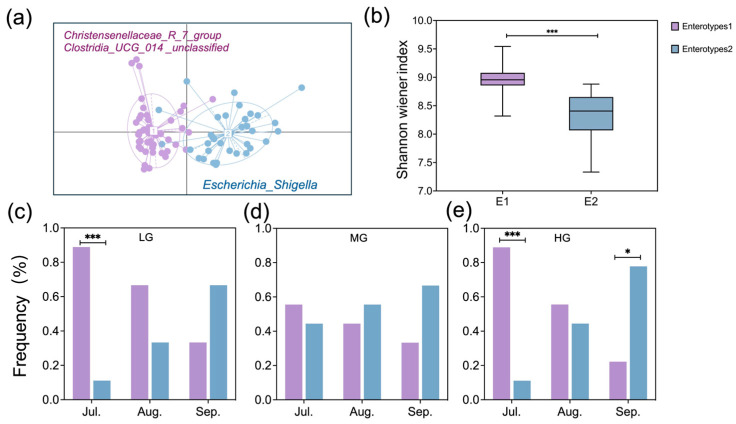
Visualization of partitioning around medoid (PAM) results in two-dimensional space for Enterotypes 1 and 2 (**a**), comparison of sheep gut microbiota Shannon–Weiner diversity between enterotypes (**b**), and the proportion of samples for each enterotype for seasonal months within LG (**c**), MG (**d**), and HG (**e**) (designations follow Figure 1). Asterisks indicate differences between enterotypes within seasonal months (* *p* < 0.05, *** *p* < 0.001).

**Figure 6 microorganisms-13-01392-f006:**
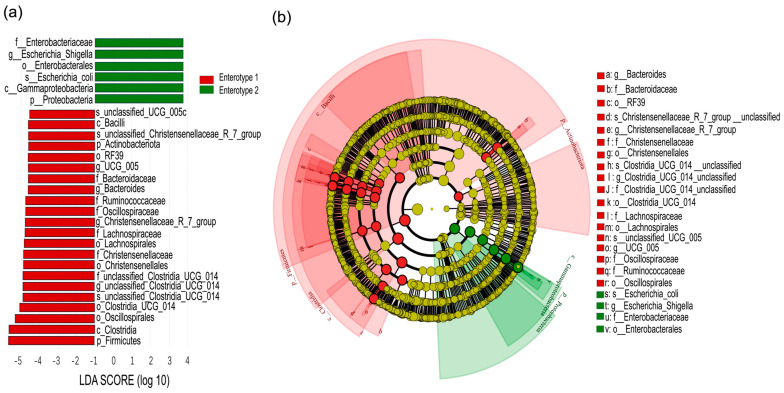
Linear discriminant analysis (LDA) effect-size scores for taxa differentially abundant between Enterotypes 1 and 2 (**a**). Cladogram generated by LDA effect size (LEfSe) analysis indicates differences in taxa abundance between enterotypes (**b**). The central dot represents the kingdom (Bacteria), and each successive ring represents the next lower phylogenetic level (phylum, class, order, family and genus). Red areas indicate taxa enriched in Enterotype 1 compared to those in the Enterotype 2, while green areas indicate taxa that were enriched in the Enterotype 2 compared to those in the Enterotype 1. Only taxa with an LDA significance threshold >3.0 are shown.

**Figure 7 microorganisms-13-01392-f007:**
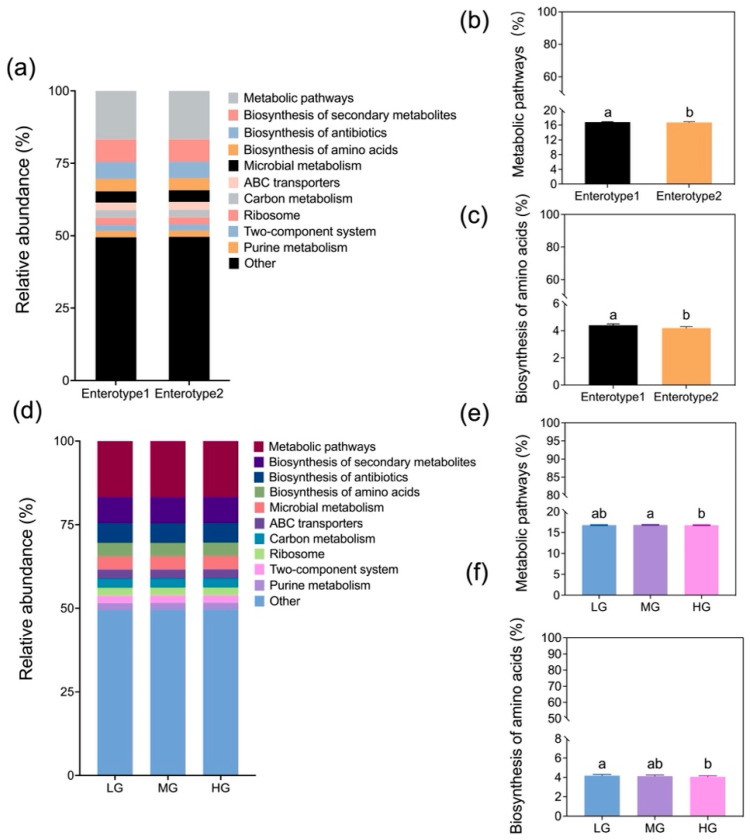
Comparison of the relative abundances for level-3 KEGG pathways between Enterotypes 1 and 2 (**a**) and among grazing levels (**d**) (designations follow Figure 1). Mean (±SE; pooled enterotypes) relative abundances for metabolic pathways and biosynthesis of amino acids by enterotypes (**b**,**c**) and grazing intensity (**e**,**f**). Means followed by the same letter are not significantly different at α = 0.05 based on Tukey’s post hoc test.

## Data Availability

The data presented in this study are openly available in [Mendeley Data] at [https://data.mendeley.com/datasets/z9ztw23dzn/1] (accessed on 3 May 2025).

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
