# Peer review of "Grazing Intensities Regulated the Effects of Seasonal Dietary Pattern on Gut Bacterial Community Composition of Sheep"

_microorganisms, 2025, doi:10.3390/microorganisms13061392_

Round 1
Reviewer 1 Report
Comments and Suggestions for Authors - Authors should briefly present the methods used to obtain the results. I suggest that the abstract begins with the research objective so that the number of words allocated to the abstract is not exceeded with the inclusion of this item. - The words contained in the title should not be repeated in the keywords. Correct Lines 72 - 77 - Authors should rewrite the sentences as a general objective. The way presented is seen as an initial presentation of the results and this is not relevant here in this item. Lines 79 - 82 - Remove. This is an expected result and should not be present here. The way presented is as if the introduction were a justification for starting a research project. Sentences should not begin with acronyms. Check this throughout the text. Lines 124 - 127 - The methods should be better described. Were the parameters analyzed presented based on dry matter? How was dry matter estimated? Present the values. Conflicts of Interest: Authors must indicate if there are any conflicts of interest. Comments on the Quality of English LanguageThe English could be improved to more clearly express the research.
Author Response
Dear editors and reviewers:
Thank you very much for reviewing the manuscript entitled " Grazing intensities regulated the effects of seasonal dietary pattern on gut bacterial community composition of sheep". Comments provided by the reviewers were helpful to improve our manuscript. The following is a list of responses to the comments:
Comments of Reviewer #1:
Authors should briefly present the methods used to obtain the results. I suggest that the abstract begins with the research objective so that the number of words allocated to the abstract is not exceeded with the inclusion of this item.
We made this correction according to your suggestion.
The words contained in the title should not be repeated in the keywords.
It was corrected as your suggestion.
Lines 72 - 77 Authors should rewrite the sentences as a general objective. The way presented is seen as an initial presentation of the results and this is not relevant here in this item.
We made this correction according to your suggestion.
Lines 79 - 82 - Remove. This is an expected result and should not be present here. The way presented is as if the introduction were a justification for starting a research project.
This was removed according to your suggestion.
Sentences should not begin with acronyms. Check this throughout the text.
This was corrected according to your suggestion.
Lines 124 - 127 - The methods should be better described. Were the parameters analyzed presented based on dry matter? How was dry matter estimated? Present the values.
The vegetation biomass and forage nutritional quality were both determined based on dry matter. This information has now been added to the revised manuscript.
Conflicts of Interest: Authors must indicate if there are any conflicts of interest.
Thank you for your suggestion. The statement regarding conflicts of interest has been added to the revised manuscript.

Reviewer 2 Report
Comments and Suggestions for Authors
The authors examined the availability, nutrient composition and diet selectivity by sheep, in summer pastures in inner Mongolia, and related these to the bacterial community of the sheep “gut”. The work is interesting in that the authors used abundance of plant group-specific DNA sequences in feces as a proxy for dietary consumption (an approach that is new to this reviewer). By establishing different intensities of grazing, and by breaking the experiment into three monthly periods, the authors provide an opportunity to examine how shifts in forage availability and composition affect the bacterial community.
As interesting as the work is, the manuscript is not without problems. In the title and the entire manuscript, the authors misuse the term “gut”. Because they are analyzing the microbial community in feces, the authors should explicitly use the term “hindgut”, and even there they need to better describe the collection methods (see specific comments below). The authors also need to include in their Discussion that the hindgut microbial community contributes far less to the digestion of the forages than does the ruminal community, which was not analyzed. This is no small matter: Any attempts to relate details of dietary intake to microbial function and animal health/performance will likely be limited by the relatively minor contribution of the hindgut microbes, relative to those of the rumen. Moreover, it is not clear if forage quality (i.e., the availability of nutrients) is as important to the hindgut microbiota, as most of the fermentable nutrients in the forage are degraded in the rumen, leaving less digestible material for the hindgut microbes. Again, this is something the authors need to point out.
The reviewer also has some major concern regarding the nutrient composition analysis (see comment to L255-261 below).
Specific comments:
Title: As noted above, the authors should use the term “hindgut” rather than “gut”. Moreover, because they examined only the bacterial community (rather than archaea, fungi and protists), the authors should substitute “bacterial community” for “microbiome”, here in the title, in the abstract, and at appropriate places in the rest of the text.
L16-17 and L106-108: How does “year” fit into measurement of grazing intensity? At LG, were there 5 sheep per ha during the experimental period (3 months, 15 Jun-15 Sept), and thus 1.25 sheep per ha for this period (i.e., 1.25 sheep/ha/0.25 yr)?. If the latter, and the plots were 2 ha in size, wouldn’t this translate to 2.5 sheep during the experimental period?
L30: Seasonal dietary what?
L43-44: This would be true only if the connection of specific enterotypes to specific health outcomes can be established.
L123: What type of grinder was used?
L126: Were a-amylase and sodium sulfite added to the ND solution?
L129-130: How were lambs ”sampled”? Was feces collected directly from the rectum via digital stimulation? Or was feces collected after it dropped to the ground?
L134: This application of fecal DNA analysis to quantify plant species is new to this reviewer, and it sound like the method has tremendous potential. Could the authors provide a few literature citations for this approach?
L179-186: Were singletons removed from the analysis?
L255-261: Something seems amiss here. The authors present amount and nutrient composition data for each of the forage groups. Evidently, the each plant piece in each of the samples was physically separated, identified at the family level, pooled into their particular plant group, to provide materials for the analysis (Tables S1 and S2). Even setting aside for the moment the labor involved and the loss of small fragments that inevitably would occur during cutting, there does not seem to be enough material for the analyses. For example, according to Table S1, the September sample contained 0.17 g/m2 of Liliaceae. These samples were collected from ten 50 cm x 50 cm plots (L118), so there would have been a total of 0.17 g/m2 x 0.25 m2/quadrant x 10 quadrants = 0.41 g of material. A substantial fraction of this small amount would have been lost during grinding and recovery, and from the remainder, the authors would have had to obtain CP, NDF and ADF (Table S2). This does not seem to be enough material for the analyses, based the methods employed (L125-127).
L285: It is interesting that the sheep exhibited “avoidance” of the Poaceae even though it represented 85-90% of the available biomass, and ultimately the majority of the diet.
L293-301: Over half the reads for the community were “low abundance taxa” (per the legend to Figure 4). What fraction of these were reads that were classifiable to the genus level (as opposed to classifiable only to family or higher taxon levels).
L318: Why was a medoid used, rather than a centroid?
Table S2: Suggest removing individual standard errors for each mean, and including instead a single pooled standard error for the means in each row.
Minor edits:
L26: Insert “that of” ahead of “Enterotype 2”.
L43: Change “due to” to “in the face of”.
Author Response
Dear editors and reviewers:
Thank you very much for reviewing the manuscript entitled " Grazing intensities regulated the effects of seasonal dietary pattern on gut bacterial community composition of sheep". Comments provided by the reviewers were helpful to improve our manuscript. The following is a list of responses to the comments:
Comments of Reviewer #2:
As interesting as the work is, the manuscript is not without problems. In the title and the entire manuscript, the authors misuse the term “gut”. Because they are analyzing the microbial community in feces, the authors should explicitly use the term “hindgut”, and even there they need to better describe the collection methods (see specific comments below).
Specific comments:
Title: As noted above, the authors should use the term “hindgut” rather than “gut”. Moreover, because they examined only the bacterial community (rather than archaea, fungi and protists), the authors should substitute “bacterial community” for “microbiome”, here in the title, in the abstract, and at appropriate places in the rest of the text.
We appreciate the reviewer’s constructive comments regarding the terminology used in our manuscript. We fully acknowledge that the microbial data in our study were obtained from fecal samples, which mainly represent the hindgut microbial community, and we agree that “hindgut” is a more anatomically precise term, however,we respectfully note that the term “gut microbiota” is widely adopted in published literature—even in studies that rely on fecal sampling and do not analyze the entire gastrointestinal tract. Our use of “gut” aligns with this convention and aims to maintain consistency with previous research. For example:
- Guo, N.; Wu, Q.; Shi, F.; Niu, J.; Zhang, T.; Degen, A.A.; Fang, Q.; Ding, L.; Shang, Z.; Zhang, Z.; et al. Seasonal dynamics of diet–gut microbiota interaction in adaptation of Yaks to life at high a npj Biofilms Microbiomes 2021, 7, 38, https://doi:10.1038/s41522-021-00207-6
- Couch, C.E.; Stagaman, K.; Spaan, R.S.; Combrink, H.J.; Sharpton, T.J.; Beechler, B.R.; Jolles, A.E. Diet and gut microbiome enterotype are associated at the population level in African Buffalo. Nat Commun 2021, 12, 2267. https://doi.org/10.1038/s41467-021-22510-8
- Ren, T.; Boutin, S.; Humphries, M.M.; Dantzer, B.; Gorrell, J.C.; Coltman, D.W.; McAdam, A.G.; Wu, M. Seasonal, spatial, and maternal effects on gut microbiome in wild red squirrels. Microbiome 2017, 5, 163. https://doi.org/10.1186/s40168-017-0382-3
These studies, which use similar fecal sampling approaches, uniformly refer to the microbial community as the “gut microbiota.” Therefore, we have retained the term “gut” in our manuscript for consistency with the established literature, while making clear in the methods and discussion that our data reflect the hindgut microbial community. We hope this clarification is acceptable. Regarding the term “microbiome,” we have carefully revised the manuscript to replace it with “bacterial community” in the title, abstract, and throughout the text, as suggested, to accurately reflect the taxonomic scope of our analysis.
L16-17 and L106-108: How does “year” fit into measurement of grazing intensity? At LG, were there 5 sheep per ha during the experimental period (3 months, 15 Jun-15 Sept), and thus 1.25 sheep per ha for this period (i.e., 1.25 sheep/ha/0.25 yr)?. If the latter, and the plots were 2 ha in size, wouldn’t this translate to 2.5 sheep during the experimental period?
Thank you for your comment. We apologize for the confusion regarding the expression of grazing intensity. In our study, the light, moderate, and heavy grazing treatments were implemented on 2-ha plots with 6, 12, and 17 lambs, respectively. Grazing was conducted continuously during the growing season (15 June–15 September), lasting for approximately 0.25 years. To standardize grazing intensity, we measured the total liveweight of lambs in each plot at the beginning of the grazing period and at monthly intervals. These weights were then converted into sheep units based on a 50 kg adult sheep equivalent. The average stocking rate over the 3-month grazing period was calculated and expressed on an annual basis (sheep units·ha⁻¹·yr⁻¹) for consistency and comparability with other studies. Therefore, the values of 0.50, 0.96, and 1.38 sheep units·ha⁻¹·yr⁻¹ represent the annualized grazing intensities based on actual stocking during the 0.25-year experimental period. We will clarify this calculation method in the revised manuscript.
L30: Seasonal dietary what?
The expression here is inaccurate and has been modified.
L43-44: This would be true only if the connection of specific enterotypes to specific health outcomes can be established.
The original expression was inaccurate, and we have revised it to adopt a more cautious wording.
L123: What type of grinder was used?
We have added the specific model of the grinder in Section 2.3 of the revised manuscript.
L126: Were a-amylase and sodium sulfite added to the ND solution?
Yes, α-amylase and sodium sulfite were accurately used during the experiment, and this has been added to the manuscript.
L129-130: How were lambs ”sampled”? Was feces collected directly from the rectum via digital stimulation? Or was feces collected after it dropped to the ground?
We used rectal sampling to collect feces, and the relevant details have been added to the manuscript.
L134: This application of fecal DNA analysis to quantify plant species is new to this reviewer, and it sound like the method has tremendous potential. Could the authors provide a few literature citations for this approach?
We have added relevant references to Section 2.4 Diet composition-DNA metabarcoding sequence analysis.
L179-186: Were singletons removed from the analysis?
Singleton ASVs were removed after the denoising step as part of the quality filtering process. We have now clarified this in the revised manuscript.
L255-261: Something seems amiss here. The authors present amount and nutrient composition data for each of the forage groups. Evidently, each plant piece in each of the samples was physically separated, identified at the family level, pooled into their particular plant group, to provide materials for the analysis (Tables S1 and S2). Even setting aside for the moment the labor involved and the loss of small fragments that inevitably would occur during cutting, there does not seem to be enough material for the analyses. For example, according to Table S1, the September sample contained 0.17 g/m2 of Liliaceae. These samples were collected from ten 50 cm x 50 cm plots (L118), so there would have been a total of 0.17 g/m2 x 0.25 m2/quadrant x 10 quadrants = 0.41 g of material. A substantial fraction of this small amount would have been lost during grinding and recovery, and from the remainder, the authors would have had to obtain CP, NDF and ADF (Table S2). This does not seem to be enough material for the analyses, based the methods employed (L125-127).
We sincerely thank the reviewer for the careful and thoughtful comment. As noted, it is indeed challenging to obtain sufficient material for nutrient analyses from small biomass samples. During the vegetation survey, we were aware of this limitation. Therefore, while biomass estimation was based on data from ten 50 cm × 50 cm quadrats per plot, we collected additional plant material when necessary. Specifically, if the biomass of a given plant group was insufficient for forage quality analysis, we established additional quadrats within each experimental plot to ensure sufficient sample collection for determining CP, NDF, and ADF contents. We have now clarified this in the revised manuscript.
L285: It is interesting that the sheep exhibited “avoidance” of the Poaceae even though it represented 85-90% of the available biomass, and ultimately the majority of the diet.
Based on your suggestion, we elaborated further on this aspect in the Discussion section.
L293-301: Over half the reads for the community were “low abundance taxa” (per the legend to Figure 4). What fraction of these were reads that were classifiable to the genus level (as opposed to classifiable only to family or higher taxon levels).
As indicated in the legend of Figure 4, we have already specified the proportion of low abundance taxa that could be classified to the genus level, as well as those that could only be classified to the family level or higher taxonomic ranks.
L318: Why was a medoid used, rather than a centroid?
We used Partitioning Around Medoids (PAM) based on Bray-Curtis dissimilarity to identify enterotypes. PAM was chosen because it selects real samples as cluster centers (medoids), making results easier to interpret for microbiome data. For validation, we also tested K-means clustering and found consistent results. Cluster quality was confirmed using the Calinski-Harabasz index and silhouette score. Key genera driving enterotype differences were identified with the SIMPER method.
Table S2: Suggest removing individual standard errors for each mean, and including instead a single pooled standard error for the means in each row.
Following your suggestion, we have revised Supplementary Table S2 accordingly.
Minor edits:
L26: Insert “that of” ahead of “Enterotype 2”.
It was corrected as your suggestion.
L43: Change “due to” to “in the face of”.
It was corrected as your suggestion.

Round 2
Reviewer 2 Report
Comments and Suggestions for Authors
The authors have addressed the reviewer’s previous concerns. Two issues remain to be addressed. First, the reviewer is satisfied that the authors acquired enough material from the less abundant plant groups to permit their chemical analysis. But the authors need to state in the manuscript what they stated in their response to reviewer, i.e., that additional sampling beyond the additional 10 quadrats was necessary to obtain additional biomass. Second, the authors noted that there was a shift from preference to avoidance of certain plant groups over the course of the season (L292-295), but they interpret shifts largely within the context of nutrient supply. (L378-385). It is perhaps worth mentioning that at least some avoidance may be due to shifting levels of plant toxins or other antinutritive compounds over the course of the grazing season.
